# Aerosol Inhalation Delivery of Ceftriaxone in Mice: Generation Procedure, Pharmacokinetics, and Therapeutic Outcome

**DOI:** 10.3390/antibiotics11101305

**Published:** 2022-09-25

**Authors:** Sergey V. Valiulin, Andrei A. Onischuk, Anatoly M. Baklanov, Sergey V. An’kov, Sergey N. Dubtsov, Alexander A. Alekseev, Nikolay N. Shkil, Ekaterina V. Nefedova, Maria E. Plokhotnichenko, Tatyana G. Tolstikova, Arseniy M. Dolgov, Galina G. Dultseva

**Affiliations:** 1Voevodsky Institute of Chemical Kinetics and Combustion SB RAS, 3 Institutskaya Str., 630090 Novosibirsk, Russia; 2Vorozhtsov Institute of Organic Chemistry SB RAS, 9 Lavrentyev Ave., 630090 Novosibirsk, Russia; 3Siberian Federal Scientific Center of Agro-BioTechnologies RAS, 630501 Krasnoobsk, Russia

**Keywords:** antibacterial agents, ceftriaxone, inhalation, mice, survival rate, *Klebsiella*, *Staphylococcus*

## Abstract

Aerosol inhalation delivery of ceftriaxone in mice was investigated. An ultrasonic nebulizer within the ranges of mean particle diameter 0.5–1.5 μm and mass concentration 0.01–0.6 μg/cm^3^ was used in inhalation experiments. Pharmacokinetic measurements were carried out using a nose-only chamber. Ceftriaxone concentration in blood serum and its mass in the lungs of mice were measured as a function of time using high-performance liquid chromatography. The body-delivered dose was within the range 3–5 mg/kg. The antibacterial effect of aerosolized ceftriaxone was investigated for mice infected with *Klebsiella pneumoniae* 82 and *Staphylococcus aureus* ATCC 25 953. The survival rate for infected mice after the treatment with ceftriaxone aerosol revealed the high antibacterial efficiency of this kind of treatment.

## 1. Introduction

Drug administration directly into the respiratory tract has been used in a number of therapeutic areas. This way of drug delivery is currently considered as an efficient tool for the treatment of both respiratory and systemic diseases [1,2,3,4]. The aerosol delivery for systemic targeting has advantages with respect to per oral treatment due to the possibility to avoid losses in the gastrointestinal tract. In addition, inhalation therapy is noninvasive and, therefore, there is no risk of needle injuries, and no need for health-care staff, in contrast to injection therapy. On the other hand, inhaled antibiotics have been used to treat respiratory infections. The rationale for use of inhaled antimicrobial agents is the higher concentrations of antibiotics achieved in the lungs compared with parenteral delivery, and reduced side effects [4,5,6,7,8,9]. The list of antibiotics marketed for inhalation includes tobramycin, colistin (nebulized and dry powder forms), nebulized aztreonam, etc. However, these drugs have been approved only for patients with cystic fibrosis. The list of drugs for inhalation has been expanded to include gentamicin, tobramycin, amikacin, ceftazidime, and amphotericin as “off-label” inhalable formulations to treat post-transplant airway infections, ventilator-associated pneumonia, bronchiectasis, and drug-resistant nontuberculous mycobacterial infections. It is generally stated that future inhaled antibiotic trials are to involve expanded disease areas [7].

In our previous works, we have elaborated the evaporation–nucleation route of aerosol generation [10,11] and used this approach to study the aerosol delivery of nonsteroid anti-inflammatory [12,13,14], hypotensive [15], and antituberculous drugs [16]. Later, we elaborated the ultrasonic technique for aerosol administration of antiviral [17] and antibiotic [18] drugs. In this paper, we apply ultrasonic aerosol generation to study inhalation delivery and antibacterial efficiency of ceftriaxone in mice.

Ceftriaxone is a representative of cephalosporins, a group of the most widely used β-lactam antibiotics, working by inhibiting the cell wall synthesis in the bacterial organism [19]. However, resistance to β-lactams is frequently developed through the bacterial synthesis of a β-lactamase. Cephalosporins of the newer generations are known to inactivate penicillin-binding proteins on the inner membranes of bacterial cell walls, which hinders the terminal stages of cell wall assembling. So, the new generation cephalosporins are expected to bring a reduced risk of the emergence of drug resistance due to the features of their chemical structure; in addition, the second and third generation cephalosporins have been proposed for patients allergic to penicillin as therapy against life threatening infections. We chose a promising third generation cephalosporin, ceftriaxone, for which no dosage forms for inhalation delivery have been adopted to date. Ceftriaxone is on the WHO list of essential medicines to be used against microorganisms that tend to be resistant to many other antibiotics. Ceftriaxone, also known in its disodium form (to enhance its water solubility) as rocephin, is used for the treatment of a number of bacterial infections, in particular middle ear infections, endocarditis, meningitis, bone and joint infections, and respiratory infections. Ceftriaxone may be given to penicillin-sensitive patients. The recommended methods of administration are intravenous and intramuscular [20]. One of the leading pathologies to be taken into account for efficient dosing of ceftriaxone was demonstrated to be pneumonia [21]. However, lung infections are on the urgency list for inhalation drug delivery directly to the site of infection. These indications, as well as the above-listed contraindications for ceftriaxone, suggest that inhalation therapy may be a promising alternative to enhance efficiency and reduce toxicity, and to eliminate most severe side effects. The goal of the present study was to develop a robust procedure for the generation of dry ceftriaxone aerosol for inhalation delivery with real-time dose control and the possibility of targeting over the respiratory tract to treat higher or lower respiratory tract infections, and to assess the therapeutic effect of the aerosolized form in laboratory animals for the models of bacterial infections of *Klebsiella pneumoniae* 82 and *Staphylococcus aureus* ATCC 25 953. It is important to find correlations between the drug concentrations in organs with particular biological effects. Therefore, pharmacokinetic measurements are given in this paper as well.

## 2. Materials and Methods

### 2.1. Aerosol Generation and Inhalation Equipment

The ceftriaxone aerosol is generated using an ultrasonic nebulizer (Figure 1). The aerosol is formed in a glass chamber 100 cm^3^ in volume, filled with an aqueous solution of ceftriaxone. The chamber is equipped with inlet and outlet tubes to blow the filtered air through them. A piezoelectric crystal incorporated into the liquid is vibrated at the frequency of 1.7 MHz, resulting in the propagation of sound waves through the medium and, as a consequence, in formation of droplets of aqueous ceftriaxone solution. These droplets are caught by the air passing through the chamber with the flow rate of 0.2 L/min. The outlet aerosol flow is then mixed with dry pure air of humidity less than 2.5%. The mixing air flow rate is 2.8 L/min. As a result, the drops are now in the dry medium, evaporating within a few milliseconds, and solid aerosol particles of ceftriaxone are formed at the outlet of the ultrasonic aerosol generator, to be supplied to the laboratory animals housed in inhalation chambers. The aerosol generation technique is described in more detail by Valiulin et al. [17,18]. In the pharmacokinetic experiments, a twelve-port nose-only (NO) inhalation chamber is used. Mice are placed radially in two tiers around the cylindrical aerosol compartment (of inner diameter 6.0 cm) so that only the nose is exposed to the aerosol. In the experiments with infected mice, a whole-body (WB) inhalation chamber made of a quartz cylinder (31.0 cm long, 2.0 L in volume) is used. In this case, the animals are immersed in the aerosol atmosphere. The ceftriaxone aerosol is supplied to the inhalation chambers with the flow rate of 3.0 L/min.

The aerosol particle size and number concentration during inhalation experiments are measured with a photoelectric (PE) counter, which has been designed and built at the Voevodsky Institute of Chemical Kinetics and Combustion, Novosibirsk, Russia. The ranges of diameters and number concentrations are 0.3–5 μm and 10^1^–5 × 10^5^ cm^−3^, respectively, for this device. The instrument works on a light scattering principle. A laser diode is used to illuminate the particles passing through the optical cell. The light scattered from each particle is collected by the receiving optics and detected by a photodetector. The photodetector converts light pulses from each particle into electrical pulses. By measuring the height of the electric signal and referencing it to the calibration curve, we determine the size of the particle, and by counting the number of pulses we determine particle number concentration. Calibration is carried out for the diameter range 0.3–4.0 μm using dibutyl phthalate (DBP) aerosol particles [22]. To generate DBP aerosol, a heterogeneous nucleation system is used consisting of a saturator, hot-wire generator of seeding particles 10 nm in diameter, and condensation chamber. The saturator creates oversaturated DBP vapor. The tungsten oxide seeding particles are mixed with DBP vapor, resulting in heterogeneous nucleation. Finally, large particles of mean diameter 0.3–4.0 μm are formed due to vapor condensation. The size spectrum of DBP particles is measured using a gravity settling technique [18]. The particle diameter is determined from the settling velocity employing the drag force equation, and using an imaging system equipped with a semiconductor laser diode and a Levenhuk C800 NG 8M digital camera. One should note that the particle size measured by the gravity settling systems is the aerodynamic diameter. The size distribution of DBP particles is well described with the log-normal function with the standard geometric deviation σ_g_ = 1.4. The calibration function is incorporated into the software of the PE counter. To demonstrate the accuracy of the calibration function, the arithmetic mean diameter determined by sedimentation technique is compared with the value determined by the PE counter (Figure 2). One can see good correspondence between these two mean diameters. Here, the mean diameters of monodispersed polystyrene latex spheres nebulized into the flow of filtered air are shown as well, demonstrating good accordance with DBP measurements. 

### 2.2. Sample Preparation and Chromatographic Analysis in Pharmacokinetic Measurements

Outbred laboratory CD-1 male mice of mass 21–25 g were used in this work. The animals for the pharmacokinetic experiments were taken from the SPF vivarium of the Federal Research Center Institute of Cytology and Genetics of the Siberian Branch of the Russian Academy of Sciences. Mice were housed in wire cages at 22–25 °C with a 12/12 h light–dark cycle. The animals had free access to standard pellet diet, tap water was available ad libitum. All the experimental procedures were approved by the Bio-Ethical Committee of the N.N. Vorozhtsov Novosibirsk Institute of Organic Chemistry of the Siberian Branch of the Russian Academy of Sciences in accordance with the European Convention for the Protection of Vertebrate Animals Used for Experimental and Other Scientific Purposes 2010, and the requirements and recommendations of the Guide for the Care and Use of Laboratory Animals.

The blood and lung samples were obtained immediately after mice were sacrificed through cervical dislocation. The blood after collecting was left to settle for 1 h, and then centrifuged for 15 min at 3000 r.p.m. The serum of volume 100 μL was mixed with acetonitrile of volume 300 μL and centrifuged at 15,000 r.p.m. Then, the sample was evaporated under a water-jet pump to a volume of 80 μL. This solution was then analyzed chromatographically. 

The lungs were initially kept in normal saline fluid for an hour. After ultrasound-assisted homogenization, the sample was centrifuged for 15 min at 15,000 r.p.m. The supernatant was frozen and kept at a temperature of −40 °C for 24 h. Then, the sample was stirred for 3 min, centrifuged once more for 15 min at 15,000 r.p.m., and the supernatant was treated with acetonitrile (1:3 by volume). Stirring and centrifuging were repeated to separate the precipitate, and the final supernatant was evaporated under the water-jet pump to a volume of about 80 μL. The volume of the sample was measured before chromatographic investigation.

Chromatographic investigation was carried out with a column filled with the reversed-phase sorbent ProntoSIL-C18. The column was thermostatted at 40 °C. Elution was carried out at a rate of 150 μL/min with a gradient of the aqueous phase (water + 1% tetrafluoroacetic acid) and the organic phase (acetonitrole + 1% tetrafluoroacetic acid), starting from acetonitrile fraction of 5% and ending with 100% CH_3_CN.

### 2.3. Histologic Analysis

Histologic analysis was performed to observe the effect of ceftriaxone aerosol on the morphology of lungs. Lungs were fixed in 4% paraformaldehyde in phosphate buffer (pH 7.2–7.4). The fixed tissues were treated in a standard way using MICROM histological equipment (Carl Zeiss), and then embedded into paraffin. Sections 3–4 µm thick were stained with hematoxylin and eosin. The slides were examined under the Axioskop 40 light microscope (Carl Zeiss).

### 2.4. Inhalation Dose

The total inhalation dose *D_T_* (μg), i.e., the total mass of particles inhaled by a mouse during the exposition, is determined as
(1)$$D_{T}=C_{A}\varepsilon v_{m}t_{0},$$
where *C_A_* (μg/cm^3^) is the particle mass concentration in the aerosol chamber, *ε* is the total lung deposition efficiency, that is, the ratio of the number of particles from the inhaled volume deposited in the respiratory tract of a mouse to the number of particles present in this volume of air in the aerosol chamber, *t*_0_ is inhalation time, *v*_*m*_ is the minute volume, i.e., the total volume inhaled by a mouse per minute. The quantity *v*_*m*_ (cm^3^/min) can be calculated as [23]
(2)$$v_{m}=595\times {\left(BW\right)}^{0.75},$$
where *BW* is the body weight of a mouse in kg. The total lung deposition efficiency is a function of the particle diameter *d*, and in the range 10 nm–10 μm it can be presented as [16,17]
(3)$$\varepsilon (d)=0.85\exp\left(-\frac{1}{2}{\left(\frac{\ln\left(d/4.0(\mathrm{nm})\right)}{2.2}\right)}^{2}\right)+0.60\exp\left(-\frac{1}{2}{\left(\frac{\ln\left(d/1590(\mathrm{nm})\right)}{1.1}\right)}^{2}\right)$$

### 2.5. Antibacterial Effect Measurements

Specific activity was studied using the archival strains of two bacterial species: *Klebsiella pneumoniae* 82 (Gram-negative species) and *Staphylococcus aureus* ATCC 25 953 (Gram-positive species). Both bacterial cultures were obtained from the archival collection of the Siberian Federal Scientific Center of Agrobiotechnologies of the Russian Academy of Sciences (SFSCA RAS).

The bacterial cultures were grown on meat-infusion agar (MIA) at a temperature of 37.5 ± 0.5 °C for 24 h, diluted in 0.85% aqueous solution of NaCl (normal saline) to the necessary concentration. Concentration was measured with a DEN-1 densitometer (Biosan). Bacterial suspensions with the concentration of 10^6^ CFU/mL were used in experiments with both *Klebsiella pneumoniae* 82 and *Staphylococcus aureus* bacterial species.

The specific effect of ceftriaxone was studied with the bacterial sepsis model in outbred CD-1 white mice of mass 21–25 g obtained from the vivarium at the Siberian Federal Scientific Center for Agrobiotechnologies, RAS. The experiment was carried out in compliance with humanity principles set out in the European Community Directive 86/609/EC (Strasbourg, 1986). Before the experiment, the animals were quarantined for 14 days under standard conditions with a light–dark mode of 12 h each, with free access to water and food; saw dust was used as bedding. The mice were infected through intraperitoneal injection of 0.5 mL of the bacterial suspension (according to the requirements described in the Methodical Guidelines for bacterial diagnostics of combined gastrointestinal infections caused by pathogenic enterobacteria in young animals, No. 13-7-2/1759 (1999), Ministry of Agriculture, RF). The animals were sacrificed through cervical dislocation, then 0.1 mL of blood was sampled from the unopened heart, placed in a Petri dish (90 mm) as a uniform layer over MIA, and incubated at 37.5 ± 0.5 °C for 24 h for the purpose of bacteriological examination. The grown colonies were counted manually.

## 3. Results and Discussion

### 3.1. Pharmacokinetics of Aerosolized Ceftriaxone

The pharmacokinetic (PK) study is an indispensable step when deriving new drug forms. It is important to describe the time course of drug concentration in the body resulting from administration of a particular dose to find correlations between drug concentrations in organs and particular biological effects. Therefore, PK measurements are considered in this subsection. 

A typical size spectrum of dried ceftriaxone aerosol particles is given in Figure 3. The frequency distribution over particle diameters is well approximated by the log-normal function *f*(*d*) with the standard geometric deviation *σ_g_* = 1.5:(4)$$f(d)=\frac{1}{\sqrt{2\pi }d\ln\sigma_{g}}\exp\left(-\frac{1}{2}\frac{{\left(\ln\frac{d}{d_{0}}\right)}^{2}}{{\left(\ln\sigma_{g}\right)}^{2}}\right),$$
where *d*_0_ is the mean geometric diameter.

The arithmetic mean diameter of ceftriaxone particles as a function of ceftriaxone concentration in nebulizing solution is shown in Figure 4. One can see that the mean diameter of dried ceftriaxone particles is proportional to the solution concentration at a power of 1/3, which means that the mean particle mass depends linearly on the solution concentration. In other words, the mean diameter of the original liquid particles generated in the nebulizer is nearly independent of the solution concentration. It is easy to calculate the mean diameter of the original liquid particles assuming that the initial ceftriaxone concentration in the drops is the same as in the nebulizing solution. Thus, using a solid ceftriaxone density equal to 2.0 g/cm^3^ (http://www.chemspider.com/Chemical-Structure.4586394.html; accessed on 23 September 2022), we have the initial diameter of the liquid drops as 3.0 ± 0.2 μm. The particle number concentration as measured by the photoelectric counter at the outlet of the aerosol generator is independent of ceftriaxone concentration in the nebulizing solution and equal to (1.8 ± 0.3) × 10^5^ cm^−3^. The aerosol mass concentration was determined by sampling with the aerosol filter (Figure 5). One can see the linear dependence of aerosol mass concentration on ceftriaxone concentration in the nebulizing solution. This linear dependence points to the independence of the initial diameter of drops from the solution concentration.

When performing the PK measurements, it is important to know the tissue responses to the drug concentration in the blood. Therefore, here we investigate the aerosol way of ceftriaxone administration in comparison with intravenous (IV) delivery. Figure 6 demonstrates the concentration of ceftriaxone in serum (a) and its mass in lungs (b) after intravenous injection. The temporal dependence of the drug concentration in serum follows the first-order kinetics with the elimination rate constant *k_e_* = 0.025 min^−1^. The mass of ceftriaxone in lungs can also be described by the exponential decay with the same first-order rate constant. Therefore, one can assume that there is an equilibrium between the mass of ceftriaxone in serum and lungs, which gives the relationship
(5)$$M_{lung}k_{lung}=Ck_{S}V_{d},$$
where *M_lung_* (μg) is the mass of ceftriaxone in lungs, *C* (μg/cm^3^) is ceftriaxone concentration in serum, *k_lung_* and *k_S_* are the first-order rate constants for forward and back reactions of the lung-to-serum transfer of ceftriaxone, *V_d_* (cm^3^) is the volume of distribution, i.e., the theoretical volume that would be necessary to contain the total amount of the administered drug at the same concentration as that observed in the blood plasma. It is easy to evaluate the equilibrium constant from the data presented in Figure 6.
(6)$$K=\frac{k_{lung}}{k_{S}V_{d}}=\frac{C}{M_{lung}}=10.6\pm 0.6\;\left({\mathrm{cm}}^{-3}\right)$$

The distribution volume in the case of IV administration can be calculated as
(7)$$V_{d}=\frac{D_{IV}}{k_{e}AUC_{IV}},$$
where *D_IV_* is the dose delivered by IV injection, $AUC_{IV}$ is the area under the curve for the temporal dependence of ceftriaxone concentration in serum in the case of IV administration. Taking into account that *D_IV_* = 5 mg/kg = 110 μg and $AUC_{IV}$ = 1680 (μg min)/cm^3^, we obtain from Equation (7) the quantity *V_d_* = 2.6 ± 0.5 cm^3^. Thus, the volume of distribution is close to the total volume of plasma per mouse (about 0.7 mL), which means that ceftriaxone is present in the blood predominantly in association with proteins [24].

Both intraperitoneal (IP) administration and aerosol inhalation delivery follow a pharmacokinetic two-compartmental model. Therefore, it is of interest to compare the time course of drug concentration in the body resulting from IP and aerosol ways of administration. Figure 7 demonstrates the concentration of ceftriaxone in serum after IP delivery. The rate of absorption from the abdominal cavity to blood (*W_P_*) can be approximated by the first-order kinetics
(8)$$W_{P}\left(t\right)=-\frac{dC_{P}}{dt}=k_{P}C_{P},$$ where *k_P_* is the first-order rate constant for ceftriaxone transfer through the peritoneal barrier, *C_P_* is the equivalent peritoneal mass concentration of ceftriaxone, *t* (min) is time. Then, the kinetic equation for ceftriaxone concentration *C*(*t*) in serum can be written as
(9)$$dC\left(t\right)/dt=k_{P}C_{P}-k_{e}C\left(t\right).$$

The joint solution of Equations (8) and (9) is
(10)$$C\left(t\right)=\frac{k_{P}C_{P}^{0}}{\left(k_{e}-k_{P}\right)}\left(\exp\left(-k_{P}t\right)-\exp\left(-(k_{e}t\right)\right),$$
where $C_{P}^{0}$ is the initial equivalent peritoneal mass concentration of ceftriaxone. To fit the experimental points presented in Figure 7 by Equation (10), the three parameters *k_P_*, *k_e_*, and $C_{P}^{0}$ are to be adjusted. The best fit values are *k_P_* = 0.035 ± 0.005 min^−1^, *k_e_* = 0.025 ± 0.005 min^−1^, and $C_{P}^{0}$ = 38 ± 4 μg/cm^3^. It is important to note that the quantity *k_e_* determined from the data on the intraperitoneal administration is in good agreement with that determined from the IV delivery (Figure 6). The distribution volume in the case of IP administration is
(11)$$V_{d}=\frac{D_{P}}{k_{e}AUC_{P}},$$
where *D_P_* is the dose delivered by IP injection, $AUC_{P}$ is the area under the curve for the temporal dependence of ceftriaxone concentration in serum in the case of IP administration. Taking into account that *D_P_* = 5 mg/kg = 110 μg and $AUC_{P}$ = 1600 (μg min)/cm^3^, we obtain from Equation (11) *V_d_* = 2.8 ± 0.5 cm^3^, which is in good agreement with the volume of distribution determined from IV administration experiments.

To determine the relationship between the aerosol delivery to the respiratory system and the response of the drug concentration in blood, the temporal dependencies of ceftriaxone concentration in serum (Figure 8a) and its mass in lungs (Figure 8b) are measured for a wide range of inhalation time intervals. In the inhalation experiment, mice were exposed to the aerosol for a definite period of time (inhalation time), then the blood and lungs were collected immediately. These measurements allowed us to determine the time of saturation of ceftriaxone in organs and find the optimal time for the aerosol exposition when studying antibacterial effect. The ceftriaxone mass in lungs is governed by the following differential equation:(12)$$\frac{dM_{lung}}{dt}=I_{lung}-k_{lung}M_{lung},$$
where *M_lung_* is the mass of ceftriaxone in lungs, *t* is inhalation time, *I_lung_* is the rate of ceftriaxone inhalation delivery to lungs, $k_{lung}$ is the apparent first-order rate constant of ceftriaxone absorption from lungs to blood. The solution of Equation (12) is
(13)$$M_{lung}=\frac{I_{lung}}{k_{lung}}\left(1-\exp\left(-k_{lung}t\right)\right).$$

The solid line in Figure 8b is the result of fitting by Equation (13). The best fit parameters are $k_{lung}$ = 0.025 ± 0.005 min^−1^, *I_lung_* = 1.0 ± 0.1 μg/min. The total rate of inhalation delivery, i.e., the total mass of particles deposited in the respiratory airways, can be evaluated as
(14)$$I_{Tot}=C_{A}\varepsilon v_{m}.$$

For the data shown in Figure 8, we have *C_A_* = 0.26 μg/cm^3^, the mean particle diameter *d* = 1.4 μm, and, as follows from Equation (3), ε ≈ 0.62. The quantity *v*_*m*_ as calculated from Equation (2) for the mean mass of a mouse of 22 g is equal to 34 cm^3^. Therefore, we obtain from Equation (14) *I_Tot_* ≈ 5.5 μg/min. The rate of total deposition can be represented as
(15)$$I_{Tot}=I_{ET}+I_{lung},$$
where *I_ET_* is the rate of deposition to the extrathoracic area, and
(16)$$\varepsilon =\varepsilon_{ET}+\varepsilon_{lung},$$
where *ε_ET_* and *ε_lung_* are the extrathoracic and lung deposition efficiencies, respectively. Taking into account that
(17)$$I_{ET}=C_{A}\varepsilon_{ET}v_{m}$$
and (18)$$I_{lung}=C_{A}\varepsilon_{lung}v_{m},$$ we obtain *ε_lung_* = 0.11 ± 0.02 and *ε_ET_* = 0.51 ± 0.04, which is in good agreement with our previous results for cefazolin particle deposition efficiencies [18].

The ceftriaxone kinetics in serum can be well described by two first-order reactions: absorption from the respiratory system to the blood systemic circulation, and elimination. The mass of ceftriaxone in the respiratory system is governed by the following differential equation:(19)$$\frac{dM_{resp}}{dt}=I_{Tot}-k_{resp}M_{resp},$$
where *M_resp_* is the mass of ceftriaxone in the whole respiratory system, $k_{resp}$ is the apparent first-order rate constant of ceftriaxone absorption from the respiratory system to blood. The ceftriaxone concentration *C*(*t*) in serum follows the equation
(20)$$\frac{dC\left(t\right)}{dt}=\frac{k_{resp}M_{resp}}{V_{d}}-C\left(t\right)k_{e}.$$

The joint solution of Equations (19) and (20) is
(21)$$C\left(t\right)=\frac{I_{Tot}}{V_{d}k_{e}}\left(1-\exp\left(-k_{e}t\right)+\frac{\exp\left(-k_{e}t\right)-\exp\left(-k_{resp}t\right)}{1-\frac{k_{resp}}{k_{e}}}\right).$$

The solid line in Figure 8a is the result of fitting by Equation (21). The best fit parameters are $k_{resp}$ = 0.050 ± 0.005 min^−1^,
*V_d_* = 3.9 ± 0.5 cm^3^.(22)

The quantity *V_d_* is higher than that obtained from IV administration experiments. This difference is probably because the inhalation dose is a few times higher than that delivered intravenously, which results in the saturation of protein binding with ceftriaxone and, as a consequence, the larger distribution volume.

As seen from Figure 8, ceftriaxone reaches its saturation in lungs and blood in about 100 min. On the other hand, in 20 min, ceftriaxone concentration in lungs is already about half of the maximum. Therefore, in the subsequent study of ceftriaxone concentration decreasing with time after aerosol administration, the inhalation time is chosen to be 20 min. As seen from Figure 9, the amount of ceftriaxone both in lungs and in serum increases monotonically with time during inhalation. After the termination of inhalation (at time *t*_0_), the mass of ceftriaxone in lungs and serum decreases with time. The dashed and solid lines in Figure 9b follow Equation (13) and exponential decay, respectively, with the rate constant $k_{lung}$ = 0.025 ± 0.005 min^−1^ in agreement with that obtained from the data shown in Figure 8b. For time *t* ≤ *t*_0_, ceftriaxone concentration in serum follows Equation (21). At *t* ≥ *t*_0_ (i.e., after the termination of inhalation) the mass of ceftriaxone in the respiratory system decreases according to the first-order law
(23)$$M_{resp}=M_{resp}^{\left(0\right)}exp\left(-k_{resp}t\right),$$
where $M_{resp}^{\left(0\right)}$ is the mass of ceftriaxone in the respiratory system at time *t*_0_. The joint solution of Equations (20) and (23) for time *t* ≥ *t*_0_ is
(24)$$C\left(t\right)=\frac{k_{resp}M_{resp}^{\left(0\right)}}{(k_{e}-k_{resp})V_{d}}\left(\exp\left(-k_{resp}(t-t_{0}\right)-\exp\left(-(k_{e}\left(t-t_{0}\right)\right)\right)+C_{0}\exp\left(-(k_{e}\left(t-t_{0}\right)\right),$$
where *C*_0_ is ceftriaxone concentration in serum just after termination of inhalation. The solid line in Figure 9a is the result of fitting by Equation (24). The best fit parameters are $k_{resp}$ = 0.050 ± 0.005 min^−1^, *k_e_* = 0.025 ± 0.005 min^−1^, and
(25)$$\frac{M_{resp}^{\left(0\right)}}{V_{d}}=6.5\pm 0.5\;\mu g/{\mathrm{cm}}^{3}.$$

The dashed line in Figure 9a follows Equation (21) with $k_{resp}$ = 0.050 ± 0.005 min^−1^, *k_e_* = 0.025 ± 0.005 min^−1^, and
(26)$$\frac{I_{Tot}}{V_{d}}=1.6\pm 0.2.$$

It is easy to calculate the total inhalation rate from Equation (14), taking into account that *C_A_* = 0.16 μg/cm^3^, ε = 0.62, and *v_m_ =* 34 cm^3^. Thus, we have $I_{Tot}$ ≈ 3.4 μg/min, and we obtain from Equation (26) *V*_d_ ≈ 2.1 ± 0.5 cm^3^. The volume of distribution as determined in the case of aerosol concentration *C_A_* = 0.16 μg/cm^3^ is less than that determined for IV and IP administration. This difference in the volume of distribution is because the inhalation dose *D_T_* = *I_Tot_* × *t*_0_ = 68 μg is about two times lower than the dose delivered by intravenous and intraperitoneal administration (110 μg). The parameters determined from the pharmacokinetic measurements are summarized in Table 1 and Table 2.

The lung histologic analysis was performed to compare representative sections from the lungs of animals treated with pure air with those of mice treated with the aerosol. In both cases, the lungs were found to have normal structures without any destructive and hemodynamic pathologic changes. The lungs showed numerous alveoli, with thin alveolar walls lined by simple squamous epithelium connected together through alveolar pores, which opened into alveolar sacs, alveolar ducts, or respiratory bronchioles. The lobar bronchi mucosa was lined by highly prismatic epithelium. The alveolar and terminal bronchial lumen did not show any emphysematous dilatation. 

### 3.2. Antibacterial Effect from Ceftriaxone Aerosol Delivery

#### 3.2.1. Experiments with *Klebsiella pneumoniae* 82

Experiments with the suspension of *Klebsiella pneumoniae* 82 were carried out with 65 animals divided into 5 groups. They are Reference 1 and Reference 2 groups of seven animals each, and three other groups for aerosol and intraperitoneal (IP) delivery with 17 animals in each of these groups, and intravenous (IV) administration of 14 animals. The animals of the Reference 1 group were not subjected to any manipulations; the animals were kept in the cage under standard conditions with the light–dark mode (day–night, 12 h each) with free access to water and food. 

The animals of the Reference 2, Aerosol, IV, and IP groups were infected. The mice of the Reference 2 group were not exposed to any manipulations after the introduction of infection. They were kept in the cage under standard conditions with the light–dark mode (day–night, 12 h each) with free access to water and food. For the animals of the Aerosol group, aerosol administration of ceftriaxone was performed in the whole-body chamber according to the scheme given in Table 3. The body delivered dose for 20 min of inhalation was *D_T_* = 4.2 ± 0.4 mg/kg. The mean particle diameter was *d* = 1.4 μm.

The Intraperitoneal and Intravenous groups were treated with ceftriaxone solution administered by injection of 0.3 mL per animal, into the abdominal cavity of the animals of the Intraperitoneal group, and into the caudal vein of the animals of the Intravenous group. The administered dose was 5 mg/kg. The manipulations with the IV and IP groups were carried out as shown in Table 4.

During the intervals between ceftriaxone administrations, the animals of Aerosol, IV, and IP groups were kept in cages under standard conditions with the light mode of day and night, 12 h each, with free access to water.

On the second day, seven mice of the Reference 2 group were dead. All the mice were surveilled, and the blood of 3 dead animals was inoculated for bacteriological examination just after expiring (to determine the bacteria, CFU/mL in the blood). In addition, animals were also drawn out from the Aerosol and IP groups (3 animals from each group) to inoculate blood for bacteriological examination. The last six animals were not taken into account in survival analysis. The results are given in Table 5.

To evaluate the survival rate, all the mice were monitored for 9 days. The animals were kept in the cages under standard conditions with the light–dark mode (day–night, 12 h each) with free access to water and food. The monitoring results are shown in Table 6.

One can see from Table 4 that the aerosol treatment demonstrated good antibacterial effect as well as IV and IP treatment. Twelve of fourteen mice of the Aerosol delivery group were alive by the end of monitoring, while all the mice from the Reference 2 group died.

#### 3.2.2. Experiments with *Staphylococcus aureus* ATCC 25 953

In the experiments with the suspension of *Staphylococcus aureus* ATCC 25 953, 40 animals were involved. The animals were divided into 4 groups, with 10 animals in each group. They included Reference 1 and Reference 2 groups, and two groups for aerosol and IP administration. 

The animals of the Reference 1 group were not subjected to any manipulations; they were kept in the cage under standard conditions with the light–dark mode (day–night, 12 h each) with free access to water and food. The animals of the three other groups were infected. The mice of the Reference 2 group were not exposed to any manipulations after the introduction of infection. They were kept in the cage under standard conditions with the light–dark mode (day–night, 12 h each) with free access to water and food. 

The animals of the Aerosol group were treated according to Table 7. The body-delivered dose for 20 min of inhalation was *D_T_* = 4.2 ± 0.4 mg/kg. The mean particle diameter was *d* = 1.4 μm.

The animals of the IP group were treated with ceftriaxone solution administered by injection of 0.3 mL per animal, into the abdominal cavity in the dose of 5 mg/kg. The manipulations with the IP group animals were as shown in Table 8.

During the intervals between aerosol and IP administrations, the animals were kept in cages under standard conditions with the light mode of day and night, 12 h each, with free access to water.

An hour after the final introduction of the medicine, 4 animals were drawn out of each group to inoculate blood for the purpose of bacteriological examination (to determine the bacteria, CFU/mL in the blood). The results are summarized in Table 9.

In each group, 6 animals were left to observe their survival for 9 days. The animals were kept in cages under standard conditions with the light–dark mode (day–night, 12 h each) with free access to water and food. The monitoring results are shown in Table 10.

One can see that five of the six infected untreated animals died, but all the mice treated by aerosol and IP administration of ceftriaxone were alive during the whole monitoring period of 9 days. Thus, the aerosol treatment demonstrated excellent antibacterial effect as well as IP treatment. 

After 9 days, to inoculate blood for the purpose of bacteriological examination, four surviving animals were drawn out of each treated group, along with the only surviving mouse from the Reference 2 group. The results are given in Table 11.

One can see from Table 5, Table 6, and Table 9, Table 10 and Table 11 that aerosol IV and IP treatments provide approximately the same reduction in the bacterial burden in mice.

## 4. Conclusions

Ceftriaxone aerosol inhalation delivery to outbred male mice has been investigated in comparison with other ways of drug delivery. For this purpose, an inhalation system including an ultrasonic aerosol generator and nose-only/whole-body inhalation chambers was used. Pharmacokinetic measurements show that inhaled ceftriaxone is accumulated in the respiratory system. The absorption to blood from the respiratory system can be described by the first-order kinetics with the rate constant *k_resp_* = 0.050 ± 0.005 min^−1^. The lung-to-blood adsorption is characterized by the first-order rate constant *k_lung_* = 0.025 ± 0.005 min^−1^, while the absorption to blood through the peritoneal barrier is characterized by the first-order rate constant *k_P_* = 0.035 ± 0.005 min^−1^. The elimination rate constant is determined from aerosol delivery experiments to be *k_e_* = 0.025 ± 0.005 min^−1^, which is in good agreement with that measured for intravenous and intraperitoneal administration. The volume of distribution is found to be dose dependent, being within the range *V_d_* ≈ 2 ÷ 4 cm^3^ for the body-delivered dose 3 ÷ 10 mg/kg. 

To investigate the antibacterial efficiency of the ceftriaxone aerosol form, outbred male mice were infected with the archival strains of *Klebsiella pneumoniae* 82 and *Staphylococcus aureus* ATCC 25 953. After the intraperitoneal injection with the concentration of 10^6^ CFU/mL, the infected animals demonstrated increased bacterial burden, however, aerosol treatment, as well as IP delivery, caused a significant reduction in the bacterial concentration in animals. Aerosol, IV, and IP treatment of infected animals resulted in approximately equal therapeutic effects. Thus, the developed ceftriaxone aerosol form is efficient against *Klebsiella pneumonia* and *Staphylococcus aureus* in mice.

The experimental data obtained in this research have confirmed the effectiveness of aerosolized antibiotics for systemic delivery. Additionally, this investigation reveals that ceftriaxone inhalation delivery results in drug accumulation in the respiratory system. Therefore, ceftriaxone pulmonary administration has a potential for the treatment of respiratory tract infections, especially multi-drug-resistant bacterial infections, as aerosol antibiotic administration can produce higher local concentrations of the drug than those obtained with conventional routes of delivery. 

## Figures and Tables

**Figure 1 antibiotics-11-01305-f001:**
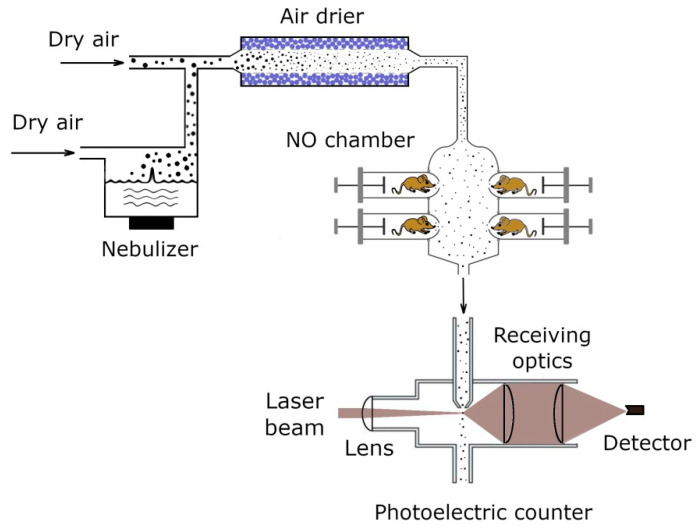
Schematic of the aerosol generation–inhalation equipment.

**Figure 2 antibiotics-11-01305-f002:**
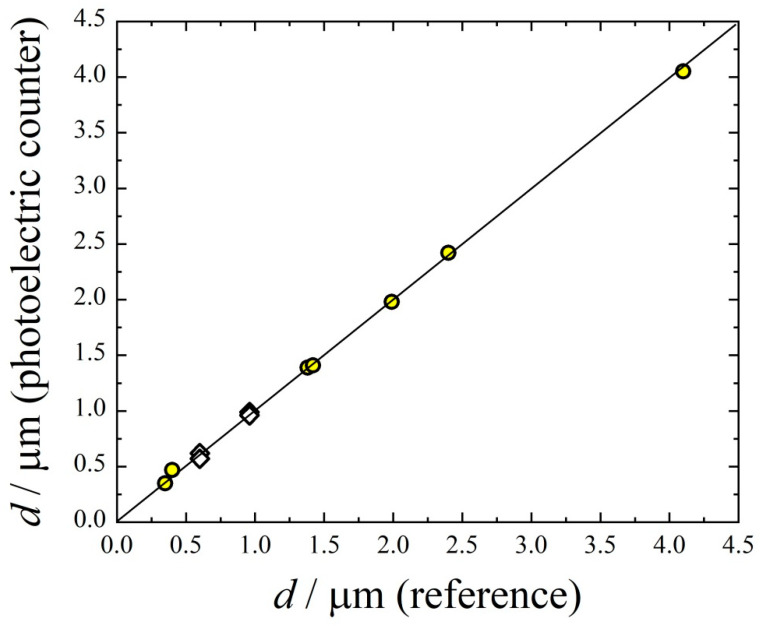
Mutual correspondence of mean diameters measured by the photoelectric counter and by the gravity settling technique (circles). Diamonds—monodispersed polystyrene latex spheres. Solid line is one-to-one correspondence.

**Figure 3 antibiotics-11-01305-f003:**
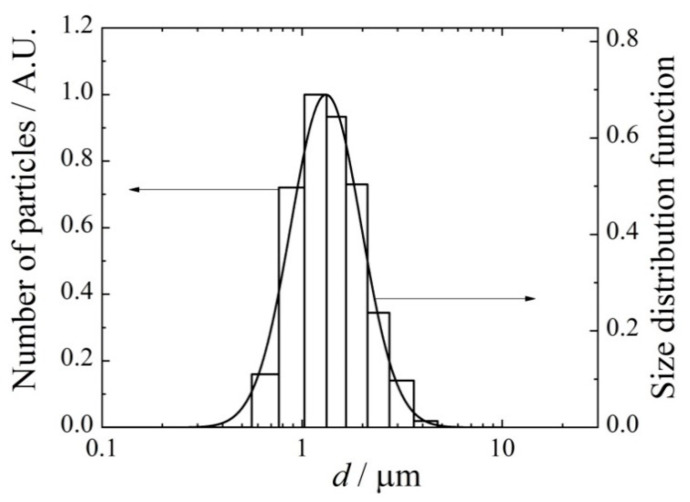
A typical size spectrum of ceftriaxone aerosol. The solid line follows Equation (4) with *σ_g_* = 1.5.

**Figure 4 antibiotics-11-01305-f004:**
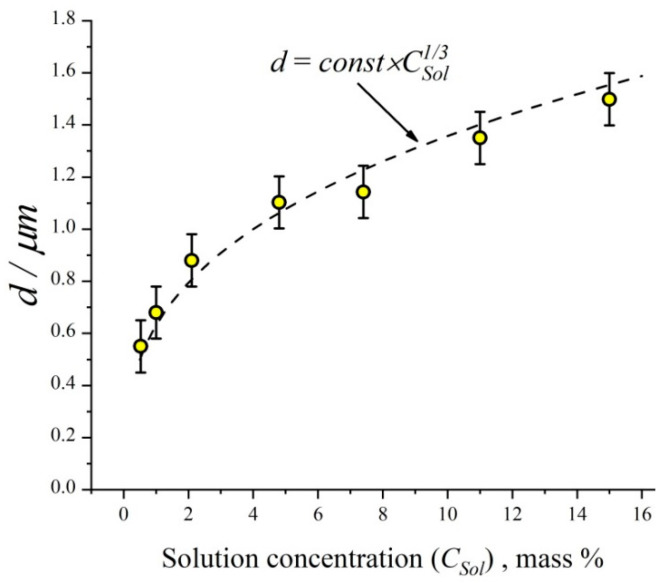
Particle arithmetic mean diameter as a function of ceftriaxone solution concentration in nebulizer. The dashed line follows the equation *d* = *const* × $C_{Sol}^{1/3}$, where *C_Sol_* is the mass concentration of ceftriaxone in the solution.

**Figure 5 antibiotics-11-01305-f005:**
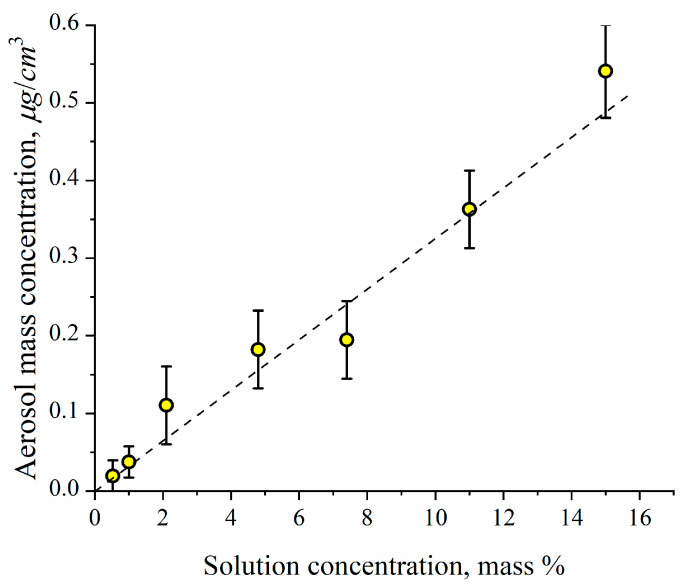
Aerosol mass concentration as measured at the outlet of aerosol generator vs. ceftriaxone concentration in the nebulizing solution. Dashed line is an eye guide.

**Figure 6 antibiotics-11-01305-f006:**
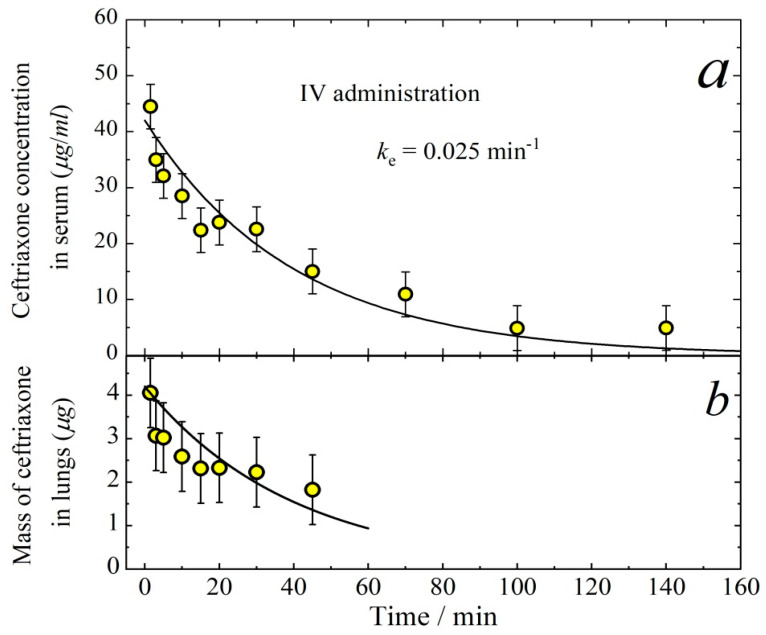
Temporal dependence of the concentration of ceftriaxone in serum (**a**) and the mass of ceftriaxone in lungs (**b**) after intravenous injection with the dose of 5 mg/kg. Solid lines follow the exponential decay ~exp(−0.025(min^−1^)·time). The time interval for the data points in panel (**b**) is limited to 50 min because at longer times the experimental error is too high with respect to the signal intensity.

**Figure 7 antibiotics-11-01305-f007:**
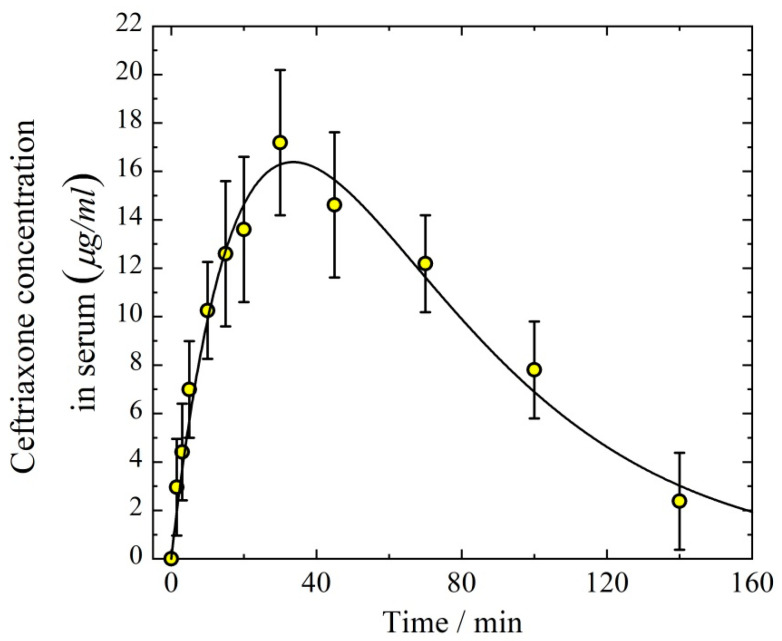
Temporal dependence of the concentration of ceftriaxone in serum after intraperitoneal injection with the dose of 5 mg/kg. The solid line follows Equation (10).

**Figure 8 antibiotics-11-01305-f008:**
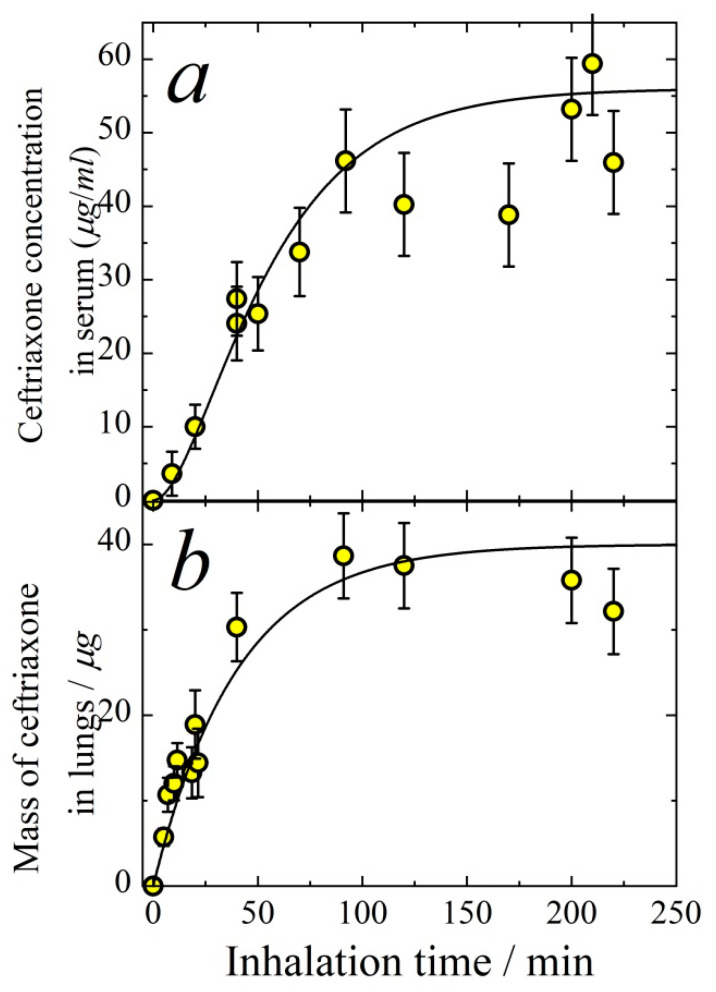
Ceftriaxone concentration in serum (**a**) and its mass in lungs (**b**) as a function of inhalation time. Aerosol mass concentration in inhalation chamber *C_A_* = 0.26 μg/cm^3^, the mean particle diameter *d* = 1.4 μm. Solid lines in panels (**a**,**b**) follow Equations (21) and (13), respectively.

**Figure 9 antibiotics-11-01305-f009:**
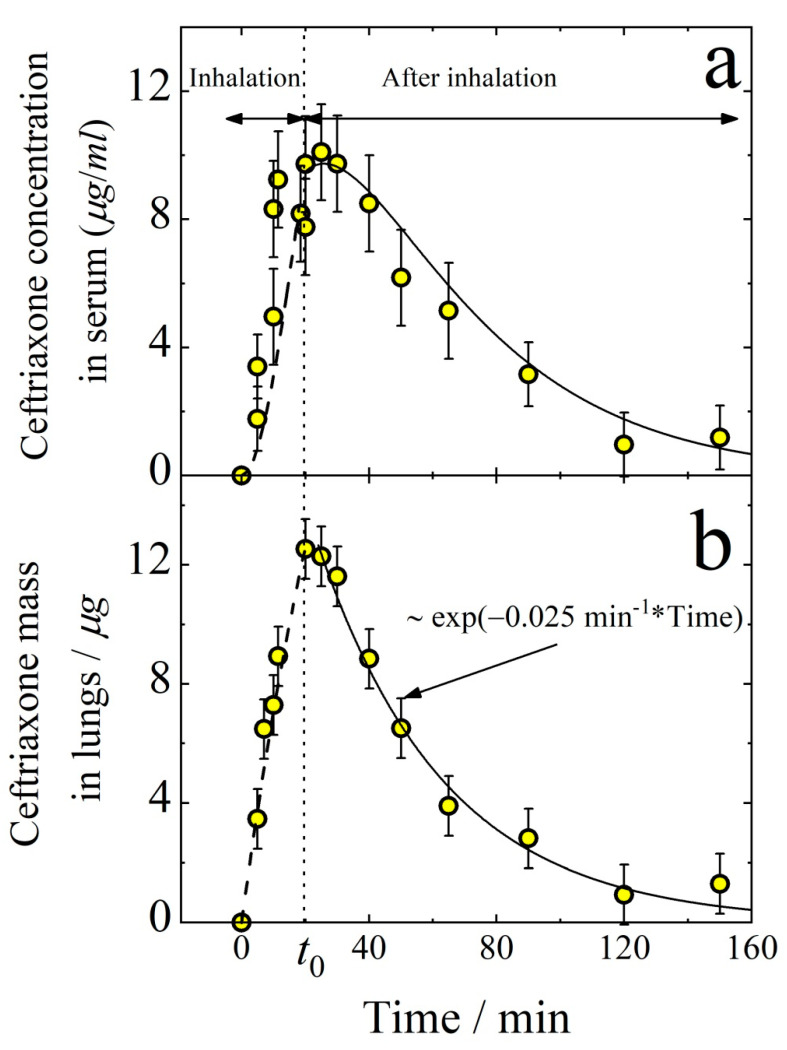
Ceftriaxone concentration in serum (**a**) and its mass in lungs (**b**) during and after inhalation. Aerosol mass concentration *C_A_* = 0.16 μg/cm^3^, the mean particle diameter *d* = 1.4 μm. Inhalation rate *I_Tot_* = 3.4 μg/min. Solid and dashed lines in panel *a* follow Equations (24) and (21), respectively, solid and dashed lines in panel *b* follow an exponential decay and Equation (13), respectively.

**Table 1 antibiotics-11-01305-t001:** Parameters determined from pharmacokinetic measurements for aerosol administration.

$C_{A}/\frac{\mu g}{cm^{3}}$	d/μm	*I_Tot_* μm/min	*I_ET_* μm/min	*I_lung_* μm/min	*k*_*lung*_ /min^−1^	*k*_*resp*_ /min^−1^	*ε*	*ε_lung_*	*ε_ET_*
0.26 ± 0.03	1.4 ± 0.2	5.5 ± 0.5	4.5 ± 0.5	1.0 ± 0.1	0.025 ± 0.005	0.050 ± 0.005	0.62 ± 0.05	0.11 ± 0.02	0.51 ± 0.04
0.16 ± 0.02	1.4 ± 0.2	3.4 ± 0.3	2.8 ± 0.3	0.60 ± 0.05	0.025 ± 0.005	0.050 ± 0.005	0.62 ± 0.05	0.11 ± 0.02	0.51 ± 0.04

**Table 2 antibiotics-11-01305-t002:** Comparison of pharmacokinetic parameters determined for different routes of drug administration.

Delivery Route	Body-Delivered Dose, mg/kg	Absorption Rate Constant, min^−1^	Elimination Rate Constant *k_e_*, min^−1^	AUC (Serum) μg·min/ cm^3^	AUC (Lungs) μg·min	Volume of Distribution *V_d_*, cm^3^
Aerosol inhalation	3.1 ± 0.3	0.050 ± 0.005	0.025 ± 0.005	760	820	2.1 ± 0.5
Intravenous	5.0 ± 0.2	-	0.025 ± 0.005	1680	-	2.6 ± 0.5
Intraperitoneal	5.0 ± 0.2	0.03 ± 0.003	0.025 ± 0.005	1600	-	2.8 ± 0.5

**Table 3 antibiotics-11-01305-t003:** Infection–treatment manipulations with mice of Aerosol group.

Time	Manipulation
0 h 0 min	Infection
0 h 10 min	20 min aerosol administration
2 h 20 min	20 min aerosol administration
4 h 40 min	20 min aerosol administration

**Table 4 antibiotics-11-01305-t004:** Infection–treatment manipulations with the mice of IV and IP groups.

Time	Manipulation
0 h 0 min	Infection
0 h 10 min	Ceftriaxone administration
2 h 0 min	Ceftriaxone administration
4 h 0 min	Ceftriaxone administration

**Table 5 antibiotics-11-01305-t005:** The data of bacteriological examination (CFU/mL) of the animals.

Mouse Number	Reference 2	Aerosol	Intraperitoneal
1	25,800	0	0
2	14,800	0	0
3	12,800	100	0

**Table 6 antibiotics-11-01305-t006:** Survival analysis of mice.

Group	Number of Animals	1st Day		2nd Day		3rd Day		4th Day		5–9th Days	
		A	D	A	D	A	D	A	D	A	D
Reference 1	7	7	0	7	0	7	0	7	0	7	0
Reference 2	10	10	0	3	7	0	10	0	10	0	10
Aerosol	14	14	0	12	2	12	2	12	2	12	2
IV	14	14	0	12	2	11	3	11	3	11	3
IP	14	14	0	13	1	13	1	13	1	13	1

*Note.* A—alive, D—dead.

**Table 7 antibiotics-11-01305-t007:** Infection–treatment manipulations with the mice of Aerosol group.

Time	Manipulation
0 h 0 min	Infection
0 h 10 min	20 min aerosol administration
2 h 20 min	20 min aerosol administration
4 h 40 min	20 min aerosol administration

**Table 8 antibiotics-11-01305-t008:** Infection–treatment manipulations with the mice of IP group.

Time	Manipulation
0 h 0 min	Infection
0 h 10 min	Ceftriaxone administration
2 h 0 min	Ceftriaxone administration
4 h 0 min	Ceftriaxone administration

**Table 9 antibiotics-11-01305-t009:** The data of bacteriological examination an hour after the final introduction of the medicine (CFU/mL).

Mouse Number	Reference 1	Reference 2	Aerosol	IP
1	0	38,500	400	400
2	0	36,000	200	600
3	0	11,700	0	0
4	0	9600	0	0

**Table 10 antibiotics-11-01305-t010:** Survival analysis of mice.

Group	Number of Animals	1st Day		2nd Day		3rd Day		4th Day		5–9th Days	
		A	D	A	D	A	D	A	D	A	D
Reference 1	6	6	0	6	0	6	0	6	0	6	0
Reference 2	6	6	0	4	2	1	5	1	5	1	5
Aerosol	6	6	0	6	0	6	0	6	0	6	0
IP	6	6	0	6	0	6	0	6	0	6	0

*Note.* A—alive, D—dead.

**Table 11 antibiotics-11-01305-t011:** The data of bacteriological examination (CFU/mL) of the animals that survived after observation for 9 days.

Mouse Number	Reference 2	Aerosol	Intraperitoneal
1	1100	100	0
2	-	200	0
3	-	0	0
4	-	0	0

## Data Availability

Data is available in this paper.

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
