# Peer review of "Aerosol Inhalation Delivery of Ceftriaxone in Mice: Generation Procedure, Pharmacokinetics, and Therapeutic Outcome"

_antibiotics, 2022, doi:10.3390/antibiotics11101305_

Round 1

Reviewer 1 Report

A very interesting paper presenting novel and sound scientific results.

However, the form and the presentation has some serious flaws and missing parts, that must be improved.

The introduction contains some obvious data for readers, these can be deleted (Lines 52-55 and Lines 57-67) and more specific info included here (e.g. https://www.jstor.org/stable/20095367).

The Materials and Methods section is the weakest part of the submission. It describes irrelevant, obvios informations (Line 86-93) but the experiments are not described at all in details! The reader finds several experiments in the paper, but Fig4, Fig5 and especially Fig6 and Fig7 results are presenting such kind of results, of which the goal and the method is not presented at all! E.g. there is an experiment presented in Fig6 and a very similar in Fig 7, what is the difference?

In antibacterial studies, the CFU/ml concentrations of the inoculum is not stated, only the volume (0.5 ml).

Serum concentrations are well presented in the Figures, but ug/ml is preferred over ug/cm3). Why not concentrations are presented in the lungs? Only ug, no ug/g is shown.

Fig 6 and Fig 7 show unbelievably high serum concentration after inhalation! How is this possible? A chicken study (https://www.jstor.org/stable/20095367) showed no detectable ceftriaxon in serum at all. With these concentrations in serum, could we even treat septic patients with inhalational ceftriaxone?

Please discuss that IP administration of ceftriaxone is biased by the fact, that the infection is also performed IP, so thus there is also a higher local concentration in the peritoneum. This might be the reason of the best results, compared to e.g. IV administration.

Why is there a difference (not stated in Materials and Methods at all) between Klebsiella and Staph infection experiments? In the latter, there is a CFU counting in the blood, in the first there is not.

In such a complex and long study, results and discussion should not be merged! The results should be discussed and compared to previous results (there are several other 3rd gen cephalosporins with results already).

Conclusion is rather an abstract, it should be a sound and short summary of the study.

Author Response

  1. The introduction contains some obvious data for readers, these can be deleted (Lines 52-55 and Lines 57-67) and more specific info included here (e.g. https://www.jstor.org/stable/20095367).

Answer: The Introduction is revised. The obvious data and more specific info are deleted. Some additional fragments are inserted (lines 60-68, page 1; 14, 15, 30-33 Page 2).

  1. The Materials and Methods section is the weakest part of the submission. It describes irrelevant, obvios informations (Line 86-93) but the experiments are not described at all in details! The reader finds several experiments in the paper, but Fig4, Fig5 and especially Fig6 and Fig7 results are presenting such kind of results, of which the goal and the method is not presented at all!

Answer: We have deleted the irrelevant, obvious information, and some details are added in accordance with the Comments of Reviewer 2 (lines 41-43 Page 2).

To give the reason for Figs. (4 - 7), the following passage is inserted in the Section "Materials and Methods" (lines 29-33 Page 6):

"The pharmacokinetic (PK) study is an indispensable step when deriving new drug forms. It is important to describe the time course of drug concentration in the body resulting from administration of a particular dose to find correlations between drug concentrations in organs and particular biological effects. Therefore, this subsection is devoted to PK measurements. "

To explain the goal of Fig.6 and Fig.7 results, the following passages are inserted:

When performing the PK measurements it is important to know the tissue responses to the drug concentration in the blood. Therefore, here we investigate the aerosol way of ceftriaxone administration in comparison with intravenous (IV) delivery (lines 20 - 22 Page 7).

Both intraperitoneal (IP) administration and aerosol inhalation delivery follow pharmacokinetic two-compartmental model. Therefore it is of interest to compare the time course of drug concentration in the body resulting from IP and aerosol ways of administration. Fig. 7 demonstrates the concentration of ceftriaxone in serum after IP delivery (lines 16 - 20 Page 9).

  1. There is an experiment presented in Fig6 and a very similar in Fig 7, what is the difference?

Answer: An additional explanation of Figs. (6, 7) is given:

"To determine the relationship between the aerosol delivery to the respiratory system and the response of the drug concentration in blood the temporal dependencies of the concentration of ceftriaxone in serum (Fig.8 a) and its mass in lungs (Fig. 8 b) are measured for a wide range of inhalation time. In the inhalation experiment, mice were exposed to the aerosol for a definite period of time (inhalation time), then the blood and lungs were collected immediately. These measurements let us determine the time of saturation of ceftriaxone in organs and find the optimal time for the aerosol exposition when studying antibacterial effect (lines 20 - 27 Page 10).

As seen from Fig. 8, ceftriaxone reaches its saturation in lungs and blood by the time of about 100 min. On the other hand, by the time 20 min the ceftriaxone concentration in lungs is already about half of maximum. Therefore, in the following study ceftriaxone concentration decrease with time after aerosol administration, the inhalation time is chosen to be 20 min (lines 6 - 10 Page 13)."

  1. In antibacterial studies, the CFU/ml concentrations of the inoculum is not stated, only the volume (0.5 ml).

Answer: In Section Antibacterial effect measurements,  the CFU/ml concentration is given, it is 106 CFU/ml:

"Concentration was measured with a DEN-1 densitometer (Biosan). Bacterial suspensions with the concentration of 106 CFU/mL were used in experiments with both Klebsiella pneumoniae 82 and Staphylococcus aureus bacterial species (lines 10 - 12 Page 6)."

  1. Serum concentrations are well presented in the Figures, but mg/ml is preferred over ug/cm3).

Answer: Serum concentration is now given in mg/ml in Figs. (6 - 9).

  1. Why not concentrations are presented in the lungs? Only mg, no mg/g is shown.

Answer: The chemical kinetic equations Eqs. (12, 13) and (19, 20) involve the mass of ceftriaxone in lungs and in respiratory system. Therefore, it is convenient to give the mass of ceftriaxone in lungs in Figs. (8, 9).

  1. Fig 6 and Fig 7 show unbelievably high serum concentration after inhalation! How is this possible? A chicken study (https://www.jstor.org/stable/20095367) showed no detectable ceftriaxone in serum at all.

Answer: The authors of the chicken study did not control the size and the aerosol concentration during the exposure. One of the possible reasons of no detectable ceftriaxone in serum may be not high enough  aerosol concentration to deliver the necessary dose. Perhaps the size of drops was too large, but the respiratory deposition efficiency decreases drastically with increasing drop diameter (d) in the range d > 2 mm.

  1. With these concentrations in serum, could we even treat septic patients with inhalational ceftriaxone?

Answer: Indeed, we showed in our previous publications that the drug concentration in serum after aerosol exposure is close to that after intravenous administration:

Onischuk AA, Tolstikova TG, Sorokina IV, Zhukova NA, Baklanov AM, Karasev VV, Dultseva GG, Boldyrev VV, Fomin VM. Anti-inflammatory effect from indomethacin nanoparticles inhaled by male mice. J Aerosol Med Pulm Drug Deliv. 2008; 21: 231–243.

Onischuk AA, Tolstikova TG, Sorokina IV, Zhukova NA, Baklanov AM, Karasev VV, Borovkova OV, Dultseva GG, Boldyrev VV, Fomin VM. Analgesic effect from ibuprofen nanoparticles inhaled by male mice. J Aerosol Med Pulm Drug Deliv. 2009; 22: 245–253.

Onischuk AA, Tolstikova TG, An’kov SV, Baklanov AM, Valiulin SV, Khvostov MV, Sorokina IV, Dultseva GG, Zhukova NA. Ibuprofen, indomethacin and diclofenac sodium nanoaerosol: generation, inhalation delivery and biological effects in mice and rats. J Aerosol Sci. 2016; 100: 164–177.

Onischuk AA, Tolstikova TG, Baklanov AM, Khvostov MV, Sorokina IV, Zhukov NA, An’kov SV, Borovkova OV, Dultseva GG, Boldyrev VV, Fomin VM, Huang GS. Generation, inhalation delivery and anti-hypertensive effect of nisoldipine nanoaerosol. J Aerosol Sci. 2014; 78: 41–54.

Valiulin SV, Onischuk AA, Baklanov AM, Dubtsov SN, An'kov SV, Tolstikova TG, Plokhotnichenko ME, Dultseva GG, Mazunina PS. Excipient-free isoniazid aerosol administration in mice: Evaporation-nucleation particle generation, pulmonary delivery and body distribution. Int J Pharm. 2019; 563: 101–109.

Valiulin SV, Onischuk AA, Dubtsov SN, Baklanov AM, An’kov SV, Plokhotnichenko ME, Tolstikova TG, Dultseva GG, Rusinov VL, Charushin VN, Fomin VM. Aerosol inhalation delivery of triazavirin in mice: outlooks for advanced therapy against novel viral infections. Journal of Pharmaceutical Sciences 2021; 110: 1316–1322.

Valiulin SV, Onischuk AA, Baklanov AM, Dubtsov SN, An’kov SV, Shkil NN, Nefedova EV,  Plokhotnichenko ME, Tolstikova TG, Dolgov AM, Dultseva GG. Aerosol inhalation delivery of cefazolin in mice: Pharmacokinetic measurements and antibacterial effect. International Journal of Pharmaceutics 2021; 607: 121013.

  1. Please discuss that IP administration of ceftriaxone is biased by the fact, that the infection is also performed IP, so thus there is also a higher local concentration in the peritoneum. This might be the reason of the best results, compared to e.g. IV administration.

Answer: The number of dead mice for the Aerosol, IV and IP deliveries was 2, 3, 1, respectively. The difference is within the experimental accuracy. Therefore, it is hardly possible to state that IP administration is more effective than the IV one.

  1. Why is there a difference (not stated in Materials and Methods at all) between Klebsiella and Staph infection experiments? In the latter, there is a CFU counting in the blood, in the first there is not.

Answer: We added the data of bacteriological examination for Klebsiella infection experiments (Table 5).

  1. In such a complex and long study, results and discussion should not be merged!

Answer: The results are divided into two parts: Pharmacokinetics and Antibacterial effect. The Antibacterial part is subdivided into two subsections: Klebsiella pneumoniae and Staphylococcus aureus infection experiments. Any additional subdivision of Results and Discussion may appear to be confusing. Therefore, we suppose it is more convenient to give the Results and Discussion merged together.

  1. The results should be discussed and compared to previous results (there are several other 3rd gen cephalosporins with results already).

Answer: Unfortunately, to our knowledge, there are only few pharmacokinetic works on aerosol delivery of 3rd generation cephalosporins available in the literature. In the Introduction, we have added a citation of the work stating that ceftazidime (one of the 3rd generation cephalosporins) is helpful as an inhalation agent: [Quon BS, Goss CH, Ramsey BW. Inhaled antibiotics for lower airway infections. Ann Am Thorac Soc. 2014 11(3):425-34], it is now ref 7 in our References. However, ceftazidime itself differs from ceftriaxone in chemical structure, water solubility, protein binding (which is for ceftazidime 5 – 22 %, typically 10 %, and for ceftriaxone it is up to 90 %); different methods were used in ceftazidime studies to generate inhalable forms (nebulized and dry powder forms were studied). We suppose that these differences make a direct comparison of the results with those of our work not very imformative. 

  1. Conclusion is rather an abstract, it should be a sound and short summary of the study.

Answer: We deleted a part of the text with unnecessary details, and added a more sound final fragment (lines 24 - 30 Page 18):

The experimental data obtained in this research have confirmed the effectiveness of aerosolized antibiotics for systemic delivery. Besides, this paper investigation reveals that ceftriaxone inhalation delivery results in the drug accumulation in the respiratory system. Therefore, ceftriaxone pulmonary administration has a potential for treatment of respiratory tract infections, especially multi-drug resistant bacterial infections, as aerosol antibiotic administration can produce higher local concentrations of a drug than those obtained with conventional routes of delivery.

Reviewer 2 Report

Overall this was an interesting and compelling study of nebulized ceftriaxone, which generated a dry powder for local treatment of lung infection. Below are some comments which may help improve the work.

There appears to be an error in Figure 2 which must be corrected, as there is a white box blocking some of the data.

An SEM image of the dried powders would be helpful to support the authors' assumption that the particles are solid spheres. In other drying processes, skin formation and collapse of particles to form "raisins" is observed, and it would be useful to rule this out.

More experimental details on the photoelectric counter should be included, as it is an important part of the analysis.

Additionally, it would be helpful to include some type of aerodynamic particle size measurement, whether by Anderson cascade impaction or next generation impactor, since aerodynamic diameter is critical to lung deposition. 

Use of the phrase "maternal solution" is confusing at least to this reviewer. Perhaps just calling it the "ceftriaxone solution" would be sufficient?

Figure 4b also has something wrong where part of the data is covered in white? Or perhaps the experiment went below the limit of quantitation at 60 min? If so, this needs to be specified in the figure.

A table of results calculated from the PK data would make a helpful summary for the reader (such as AUC, Vd, etc.)

In many systems, a substantial amount of powder is deposited onto the nose or face of the animal, where it is licked off and swallowed. This leads to significant ingestion via the GI tract, and potential for absorption. Please comment on whether this is relevant for the specific compound studied here, and if so how it would impact results. 

In equation 27 it is not necessary to write out the arithmetic.

Some explanation of the rationale for treating the animals three times after exposure, as well as the timing of the exposures would be enlightening.

A visual schematic of the droplet formation and breakup during nebulization could be helpful, even if reprinted from the authors' previous work.

Author Response

  1. There appears to be an error in Figure 2 which must be corrected, as there is a white box blocking some of the data.

Answer: The white block is deleted (now in Fig. 4)

  1. An SEM image of the dried powders would be helpful to support the authors' assumption that the particles are solid spheres. In other drying processes, skin formation and collapse of particles to form "raisins" is observed, and it would be useful to rule this out.

Answer: We agree, it would be useful to add SEM images to discuss the mechanism of particle formation. However, the paper is overloaded already, and it is not reasonable to add any SEM results. The particle diameter measured by the photoelectric counter is the aerodynamic one (see reply to the comment No 4.) and it is enough to estimate the respiratory deposition efficiency.

  1. More experimental details on the photoelectric counter should be included, as it is an important part of the analysis.

Answer: We added more details on the photoelectric (PE) counter (From line 4 Page 3 to line 7 Page 4).

  1. Additionally, it would be helpful to include some type of aerodynamic particle size measurement, whether by Anderson cascade impaction or next generation impactor, since aerodynamic diameter is critical to lung deposition. 

Answer: We added some details on the calibration of PE counter and showed that actually we measured the aerodynamic diameter (From line 4 Page 3 to line 7 Page 4).

  1. Use of the phrase "maternal solution" is confusing at least to this reviewer. Perhaps just calling it the "ceftriaxone solution" would be sufficient?

Answer: We changed "maternal solution" for "nebulizing solution".

  1. Figure 4b also has something wrong where part of the data is covered in white? Or perhaps the experiment went below the limit of quantitation at 60 min? If so, this needs to be specified in the figure.

Answer: We added to the legend of Fig 6 (previously Fig 4) the following clarification:

"The time interval for the data points in panel b is limited by 50 min because at longer times the experimental error is too high with respect to the intensity of signal."

  1. A table of results calculated from the PK data would make a helpful summary for the reader (such as AUC, Vd, etc.)

Answer: We added Tables 1, 2 with the PK data.

  1. In many systems, a substantial amount of powder is deposited onto the nose or face of the animal, where it is licked off and swallowed. This leads to significant ingestion via the GI tract, and potential for absorption. Please comment on whether this is relevant for the specific compound studied here, and if so how it would impact results. 

Answer: Ceftriaxone is known to be poorly available after oral administration [e.g. M Kawish, A Elhissi, T, Jabri et al. Pharmaceutics 2020, 12, 492; doi: 10.3390/pharmaceutics12060492]. Its oral bioavailability is hindered by its gastric instability and very low permeability through gastrointestinal epithelium, so special measures including, for example, modifications with bile salts or metal complexes are needed to improve oral bioavailability [Ba B, Gaudin K, Désiré A, et al. Antimicrob Agents Chemother. 2018;62(12):1–12. doi:10.1128/AAC.01170-18]. We use non-modified ceftriaxone in our work. So, swallowing is not expected to cause any increase in ceftriaxone concentration in blood.

  1. In equation 27 it is not necessary to write out the arithmetic.

Answer: We deleted arithmetic.

  1. Some explanation of the rationale for treating the animals three times after exposure, as well as the timing of the exposures would be enlightening.

Answer: We chose three times as the optimal number of treatments relying on the results of many studies of antibiotics over various murine infection models [for example, Sauve C, Azoulay-Dupuis E, Moine P, Darras-Joly C, Rieux V, Carbon C, Bédos JP. Efficacies of cefotaxime and ceftriaxone in a mouse model of pneumonia induced by two penicillin- and cephalosporin-resistant strains of Streptococcus pneumoniae. Antimicrob Agents Chemother. 1996;40(12):2829-34. doi: 10.1128/AAC.40.12.2829.]. A single administration is insufficient to reveal the antimicrobial effect of a drug, because the mechanism of ceftriaxone action involves interference with the microbial cell wall formation, and this process takes some time. Treatment for three times appears to be the optimal procedure [Moine P, Vallée E, Azoulay-Dupuis E, Bourget P, Bédos JP, Bauchet J, Pocidalo JJ. In vivo efficacy of a broad-spectrum cephalosporin, ceftriaxone, against penicillin-susceptible and -resistant strains of Streptococcus pneumoniae in a mouse pneumonia model. Antimicrob Agents Chemother. 1994;38(9):1953-8. doi: 10.1128/AAC.38.9.1953.]. The timing of the exposures was chosen on the basis of pharmacokinetic curves. We determined (Fig. 9) that ceftriaxone concentration in blood remained at rather significant level (not lower than 4 μg/cm3) for 60-80 min after inhalation, and the timing was chosen within this interval.

Reviewer 3 Report

The article by Sergey V. Valiulin and coworkers have reported a detailed study of the aerosol inhalation delivery of ceftriaxone. The paper outlines the study following their previous work in this area and in a stepwise manner explains all the details of the performed experiment, used mathematical formulae, deduction and finally note the observations. This is a flawless study and I recommend for publication. 

Author Response

Reviewer provided a truly encouraging comment saying that our study is flawless. We would like to thank Reviewer for having recommended our paper for publication.

Round 2

Reviewer 1 Report

The paper became much better after the corrections, I accept in this form.